# Synergistic Immunity and Protection in Mice by Co-Immunization with DNA Vaccines Encoding the Spike Protein and Other Structural Proteins of SARS-CoV-2

**DOI:** 10.3390/vaccines11020243

**Published:** 2023-01-21

**Authors:** Jinni Chen, Baoying Huang, Yao Deng, Wen Wang, Chengcheng Zhai, Di Han, Na Wang, Ying Zhao, Desheng Zhai, Wenjie Tan

**Affiliations:** 1School of Public Health, Xinxiang Medical University, Xinxiang 453003, China; 2NHC Key Laboratory of Biosafety, National Institute for Viral Disease Control and Prevention, China CDC, Beijing 102206, China; 3School of Pharmacy, Xinxiang Medical University, Xinxiang 453003, China

**Keywords:** COVID-19, SARS-CoV-2, co-immunization, DNA vaccine, spike protein, structural protein

## Abstract

The emergence of new variants of severe acute respiratory syndrome coronavirus 2 (SARS-CoV-2) has generated recurring worldwide infection outbreaks. These highly mutated variants reduce the effectiveness of current coronavirus disease 2019 (COVID-19) vaccines, which are designed to target only the spike (S) protein of the original virus. Except for the S of SARS-CoV-2, the immunoprotective potential of other structural proteins (nucleocapsid, N; envelope, E; membrane, M) as vaccine target antigens is still unclear and worthy of investigation. In this study, synthetic DNA vaccines encoding four SARS-CoV-2 structural proteins (pS, pN, pE, and pM) were developed, and mice were immunized with three doses via intramuscular injection and electroporation. Notably, co-immunization with two DNA vaccines that expressed the S and N proteins induced higher neutralizing antibodies and was more effective in reducing the SARS-CoV-2 viral load than the S protein alone in mice. In addition, pS co-immunization with either pN or pE + pM induced a higher S protein-specific cellular immunity after three immunizations and caused milder histopathological changes than pS alone post-challenge. The role of the conserved structural proteins of SARS-CoV-2, including the N/E/M proteins, should be investigated further for their applications in vaccine design, such as mRNA vaccines.

## 1. Introduction

Severe acute respiratory syndrome coronavirus 2 (SARS-CoV-2) is the cause of coronavirus disease 2019 (COVID-19), which has caused millions of infections and deaths worldwide and has jeopardized human health and the global economy. Although effective therapeutic approaches are still unavailable, they have advanced rapidly, including the application of CAR-T cell therapy and nanotechnology [1,2]. Vaccination is an effective way to control the pandemic, and several vaccines have been approved for use by various regulatory health bodies [3,4].

The coronavirus genome encodes four main structural proteins, namely, the spike (S), nucleocapsid (N), membrane (M), and envelope (E) proteins, which are responsible for virion assembly and the suppression of the host immune response [5]. The S protein is composed of 1273 amino acid residues containing two subunits, namely, S1 and S2. It mediates viral entry and is a major target for developing coronavirus vaccines [6,7,8,9,10,11]. However, the SARS-CoV-2 S protein has a high mutation frequency. Not surprisingly, in SARS-CoV-2, an RNA virus, the mutation is continuous and inevitable. There have been five SARS-CoV-2 variants of concern (VOC) that have emerged since September 2020, including B.1.1.7 (UK, Alpha), B.1.351 (South Africa, Beta), P.1 (Brazil, Gamma), B.1.617.2 (India, Delta), and B.1.1.529 (South Africa, Omicron) (Andreano and Rappuoli, 2021; Gupta, 2021). They all have several mutations in the spike protein [12]. These variants threaten the effectiveness of current COVID-19 vaccines, which are designed to target only the spike protein.

The N protein of SARS-CoV-2 binds to viral RNA through a 140-amino-acid-long RNA-binding domain in their core in a “bead on a string” manner. It is highly conserved among coronaviruses, sharing a ~90% sequence identity with that of SARS-CoV, and it is also the only structural protein inside the virion [13]. In addition, it plays an important role in packaging viral RNA into the ribonucleocapsid complex and is necessary for viral RNA replication, virion assembly, and the release from host cells [14]. Based on the high sequence similarity of the N protein in coronaviruses, it may be suggested as a cross-protection vaccine target. We previously found that co-immunization with two DNA vaccines expressing E and M proteins provides partial protection against SARS-CoV-2, and this method should be considered during vaccine development [15]. Depending on the landscape document from the WHO, there are usually seven strategies for SARS-CoV-2 vaccine candidates, which can be further divided into three categories: first, protein-based vaccines, including inactivated virus vaccines, virus-like particles, and protein subunit vaccines; second, gene-based vaccines, including virus-vectored vaccines, DNA vaccines, and mRNA vaccines; third, a combination of both protein-based and gene-based approaches, such as live-attenuated virus vaccines. DNA technologies, as novel gene-based vaccines strategies, can rapidly compare multiple vaccine candidates and strategies during preclinical testing [16,17]. Theoretically, almost all viral proteins are potential immunogens and vaccine targets. However, to the best of our knowledge, synthetic DNA vaccines’ immunogenicity and protective potential in encoding SARS-CoV-2 S proteins and other structural proteins have yet to be systematically reported.

Four DNA vaccines expressing SARS- CoV-2 S, N, E, and M l proteins were evaluated for their immunogenicity and protective efficacy in mice to explore the immunological effects of S in combination with other structural proteins.

## 2. Materials and Methods

### 2.1. Cells

Huh7.5 cells and human embryonic kidney 293T cells were cultured at 37 °C in a humidified atmosphere at 5% CO_2_ throughout the study. The cells were cultured in DMEM medium (HyClone, Logan, UT, USA), supplemented with 10% FBS (GEMINI Co., Shanghai, China) and 1% penicillin-streptomycin (Gibco, New York, NY, USA). All cell lines were confirmed negative for mycoplasma contamination.

### 2.2. Construction of DNA Vaccines Coding SARS-CoV-2 S/N/E/M

The SARS-CoV-2 S/N protein-encoding gene, containing an N-terminal Kozak sequence (GCCACC) followed by an initiation codon (ATG), was synthesized using a mammalian-optimized codon (GenScript Co., Nanjing, China). It was then cloned into the expression vector pcDNA3.1 (+) via EcoRI and XbaI digestion and named pS/pN (DNA vaccines) (Figure 1A). The pE/pM protein was constructed and identified as described previously [15]. Vaccines were prepared using endotoxin-free Maxiprep kits (Qiagen, Beijing, China), and sequences were confirmed using Sanger DNA sequencing. The expression of the S/N protein was confirmed using western blotting and anti-S (Sino Biological, Beijing, China)/anti-N antibodies diluted at 1:1000. These experiments were conducted as described previously [15,18].

### 2.3. Immunization and Challenge

Female BALB/c mice (Charles River Laboratories, France) at 6 weeks of age were housed at the National Institute of Occupational Health and Poison Control in a 21 °C and humidity-controlled environment with 12 h light/dark cycles. Meanwhile, food and water were provided ad libitum, and all animal experiments were approved by the Committee on the Ethics of Animal Experiments of the Chinese Center for Disease Control and Prevention (China CDC). The research complied with the relevant ethical regulations.

Mice were divided randomly into five groups and immunized with pS/pN alone or co-immunized with pS + pN or pS + pE + pM on days 0, 21, and 42 via intramuscular injection plus electroporation (35 mg/50 mL) (Figure 2) [19,20]. Briefly, DNA vaccines were injected into the tibialis anterior (TA) muscle and immediately pulsed with electricity using a 5 mm-apart two-needle array electrode (ECM830; BTX) with needles. Sera from the mice were collected for humoral immune response analysis, and mouse spleens were processed to measure the cellular immune response (Figure 2).

SARS-CoV-2 challenge experiments were conducted as described previously [15,21]. Briefly, the mice were anesthetized and then transduced intranasally with 2.5 × 10^8^ PFU of Ad5-hACE2 in a total volume of 45 μL. Five days post-transduction, the mice were anesthetized and then intranasally challenged with 1 × 10^5^ TCID_50_ of SARS-CoV-2 (Wuhan/IVDC-HB-02/2019) in a total volume of 50 μL of saline buffer. All of the work with live SARS-CoV-2 in mouse models was performed in Animal Biosafety Level 3 (ABSL-3) laboratories.

### 2.4. Enzyme-Linked Immunosorbent Assay

Enzyme-linked immunosorbent assays (ELISA) were conducted as described previously [15]. Briefly, S (purchased from Sino Biological)/N proteins (a gift from Song) diluted in carbonate buffer (0.1 M, pH 9.6) were coated onto 96-well EIA/RIA plates (Thermo Fisher Scientific, Waltham, MA, USA) overnight at 4 °C. The plates were blocked with 200 μL of 10% goat serum in PBS at 37 °C for 2 h, followed by washing five times with PBST. Then, serum samples serially diluted in 2% goat serum in PBS were added and incubated for 2 h at 37 °C, followed by five washes with PBST. HRP-conjugated goat anti-mouse IgG Ab (1:5000) was added at 37 °C for 1 h. A total of 100 μL of TMB substrate was added to each well and quenched with 50 μL of 2M H_2_SO_4_. The absorbance was read at a wavelength of 450 nm using SPECTR Ostar Nano (BIO-GENE, Hong Kong, China).

### 2.5. Pseudovirus Infection and Neutralization Experiments

The pseudovirus neutralization assay was performed as described previously [21,22]. A plasmid expressing the ancestral virus S protein was previously constructed [22]. The Omicron variant SARS-CoV-2 spike protein gene (GISAID: EPI_ISL_6590782.2) was synthesized (a gift from Vazyme Biotech Co., Ltd., Nanjing, China) using a mammalian-optimized codon and cloned into the pcDNA3.1 vector, as described previously [22]. Briefly, plasmids expressing a luciferase reporter and plasmids expressing the S protein were co-transfected into HEK 293T cells using the X-treme GENE HP DNA Transfection Reagent. The cell culture was refreshed 6 h after transfection, and the pseudovirus-containing supernatant was harvested after 48 h and stored at −70 °C. In the pseudovirus neutralization assay, an equal volume of the pseudovirus-containing supernatant was then added to the diluted serum. The serum–virus mixture was then incubated at 37 °C for 1 h. The Huh7.5 cells culture media were then replaced with 100 µL of the serum–virus mixture and were incubated at 37 °C for 12 h. Cells cultured with only SARS-CoV-2 pseudoviruses were run in parallel. The media were then replaced with DMEM (2% FBS), and the incubation was incubated at 37 °C for 48 h. Then, the luciferase signal was measured using the Bright-Glo firefly luciferase kit (Promega).

### 2.6. SARS-CoV-2 Neutralization Assay

SARS-CoV-2 (Wuhan/IVDC-HB-02/2019) was used in this experiment. Briefly, the sera were diluted twofold from a starting dilution of 1:10, mixed with an equal volume (10–15 pfu/well) of live SARS-CoV-2, and incubated for 1 h at 37 °C, after which they were added to the seeded Vero cells. After incubation at 37 °C for 48 h, a cytopathic effect (CPE) was observed, and 100 μL of the culture supernatant was harvested for nucleic acid extraction and real-time fluorescence reverse transcription PCR (RT-PCR). The median neutralization dose (ND50) was calculated using the Reed–Munch method [15].

### 2.7. IFN-γ ELISpot Assay

The peptide pools spanning the entire S/N/E/M protein as consecutive 15-mers overlapping by 10 amino acids were synthesized by Scilight Biotechnology, LLC. Approximately 2.5 mg of each purified peptide in the peptide pool was present per vial. The experiment was conducted as described previously [18]. Briefly, 96-well plates (BD ELISPOT Set, USA) were coated with anti–IFN-γ capture Ab and incubated overnight at 4 °C. The plates were blocked with the complete culture medium after washing three times. Splenocytes were harvested after the mice were euthanized on days 35 and 120 Fresh single-cell suspensions from each group were plated at 5 × 106 per well, and peptides were added. The plates were then incubated at 37 °C in 5% CO_2_ for 22 h and detected using an ELISpot plate reader (Biosys, So. Pasadena, CA, USA). A spot-forming unit (SFU) represents a T cell-secreting IFN-γ.

### 2.8. Evaluation of Protection in Mice Post-SARS-CoV-2 Challenge

The experiments were conducted as described previously [15,21]. Briefly, the lungs were harvested after the mice were euthanized. Half of the tissues were used for nucleic acid extraction, real-time fluorescence RT-PCR, and TCID50. The other half were sent to the College of Veterinary Medicine, China Agricultural University for pathological evaluation.

### 2.9. Statistical Analysis

Unpaired *t*-tests, two-way ANOVA tests, and Dunnett’s multiple comparisons test were performed using GraphPad Prism 7.0 (GraphPad Software LLC). *p*-values < 0.05 were considered statistically significant (* *p* < 0.05; ** *p* < 0.01; *** *p* < 0.001; **** *p* < 0.0001).

## 3. Results

### 3.1. Characterization of DNA Vaccines

E and M protein levels were detected using western blotting. We measured the expression of the encoded S/N/E/M proteins of SARS-CoV-2 in HEK-293T cells transfected with pS/pN/pE/pM plasmids via western blot analysis, using anti-S/anti-N antibodies and an anti-6 x His antibody, in the cell lysates. The bands approximated the predicted molecular weight of the S (140–142 kDa), N (45 kDa), E (10 kDa), and M (22–25 kDa) proteins (Figure 1B).

### 3.2. Robust and Sustained Anti-S and/or anti-N IgG Production Induced by pS and/or pN DNA Vaccines

Serum was collected from BALB/c mice at 35, 56, 96, and 120 days (Figure 2). Anti-S/anti-N IgG levels were detected using ELISA. The magnitude of the S- or N-specific IgG response induced by pS or pN was increased in the serum following the first and second boosts. The anti-S and anti-N IgG titers were higher in the pS + pN group than they were in the other groups; however, the difference was not statistically significant (Figure 3A,B). No robust E/M protein-specific antibody responses were detected, which is consistent with the results of a previous study (data not shown) [15].

### 3.3. High Levels of Neutralizing Antibody Induced by Co-Immunization with pS and pN Vaccines

The neutralizing titers of serially diluted serum samples were determined using the pseudotyped SARS-CoV-2 virus. The highest levels of neutralizing antibodies (nAbs) were observed in the pS + pN group, with the reciprocal EC50 geometric mean titers reaching 2988 (on day 35) and 3578 (on day 56) (Figure 3C). Similar results were observed using the live virus microneutralization (MN) assay, wherein the levels of nAbs in the pS + pN group were higher than those in the S group on days 56 and 96 (*p* < 0.05; Figure 3D). Moreover, the levels of nAbs in the pS + pN group on day 56 (second boost) were significantly higher than those on day 35 (*p* < 0.05; Figure 3D).

The neutralizing activity of each vaccine regimen against the SARS-CoV-2 Omicron variant was further determined using the pseudotyped platform and serum samples. The neutralization profile against the Omicron virus on days 35 and 56 was similar to that against the ancestral virus (Figure 3E), suggesting that the pS + pN treatment had cross-neutralizing potency.

### 3.4. T-Cell Responses Induced by DNA Vaccination

As previously described, the T-cell responses against the SARS-CoV-2 S/N/E/M antigens were estimated using IFN-γ ELISpot, as previously described [15]. As expected, both the pS + pN and pS + pE + pM regimens induced significantly higher levels of IFNγ+ T cells specific for the S protein on day 120 than on day 35 (*p* < 0.05; Figure 4A). Moreover, the number of IFNγ+ T cells specific for the N protein on day 120 (second boost) was significantly higher than that on day 35 in the pS + pN group (*p* < 0.05; Figure 4B). Finally, the number of IFNγ+ T cells specific for the M protein on day 120 (second boost) was significantly higher than that on day 35 in both groups (*p* < 0.05; Figure 4D).

### 3.5. Synergistic Protection Induced by Co-Immunization with pS/pN or pS/pE/pM

We then evaluated the protective efficacy of DNA vaccines using hACE2 mice immunized post-challenge with the ancestral SARS-CoV-2 virus. Following the challenge, the mice in the mock group exhibited gradual weight loss. In contrast, the mice immunized with either pS or pS+ showed mild weight loss immediately after infection, followed by recovery (Figure 5A). No live virus was detected in the mice vaccinated with pS, pS + pN or pS + pE + pM. Furthermore, the pS + pN vaccination significantly reduced the viral RNA copy numbers compared to those obtained with pS vaccination alone (*p* = 0.0228; Figure 5B). Moreover, lung histopathology demonstrated that mice in both the mock and pN groups showed focal patches of inflammation, pleural invagination, alveolar collapse, high levels of inflammatory cell infiltration, and hemorrhagic areas. In comparison, mice treated with either pS + pN or pS + pE + pM exhibited milder histopathological changes and lower INHAND scores post-challenge than the other group (Figure 5C).

## 4. Discussion

In this study, co-immunization with two DNA vaccines expressing the S and N proteins induced high levels of nAbs and was highly effective in reducing the SARS-CoV-2 viral load in mice. DNA vaccines expressing the S protein induced increased S protein-specific cellular immunity levels after three immunizations when mice were co-immunized with N/E and M proteins and alleviated the histopathological changes post-challenge. To the best of our knowledge, this is the first report revealing the synergistic enhancement of immunity and protection in mice using a DNA vaccine encoding the S protein when they are co-immunized with DNA vaccines encoding other structural proteins of SARS-CoV-2.

Immunodominant B-cell epitopes in N antigen regions have been observed in several studies. N-based vaccines usually cannot induce nAbs, likely because the N protein is not displayed on the viral surface. Notably, co-immunization with the S and N proteins induced higher levels of nAbs against the ancestral and Omicron SARS-CoV-2 virus than the other groups. Increased nAb responses are associated with a better viral clearance and protective efficacy. Our results demonstrated that the pS + pN treatment was more effective than the pS treatment alone in reducing the SARS-CoV-2 viral load post-challenge. A previous study reported that hamsters immunized with a vaccine co-expressing the M and N proteins were protected against severe weight loss and lung pathology and had significantly reduced viral titers in the oropharynx and lungs post-SARS-CoV-2 challenge, which is consistent with our results [23]. Unfortunately, the reduction in virus titers cannot be specifically attributed to the M or N protein, and nAb levels were not evaluated in this study. One SARS-CoV-2 mRNA vaccine study reported that S + N co-immunization induced an augmented S-specific CD8+ T-cell response and neutralizing antibodies activity, providing better protection in the lungs against the Delta variant compared with S alone, which is consistent with the results of this study [24]. Another study reported that the N protein of transmissible gastroenteritis coronavirus promoted the synthesis of neutralizing antibodies when porcine TGEV-IMMUNE cells were stimulated with a combination of S and N proteins in vitro, and this effect might be explained by the helper T-lymphocyte response to the N protein [25].

Immunodominant CD4+/CD8+ T-cell epitopes in N-antigen regions have been identified previously. Several studies have reported that vaccines based on the SARS-CoV-2 N protein effectively induce cellular immune responses. The S + N group showed increased S protein-specific cellular immunity levels after three immunizations. One SARS-CoV-2 mRNA vaccine study reported that combinatorial S + N induced an augmented S-specific CD8+ T-cell response compared with S alone, which is consistent with our results [24]. Another study reported that T-cell responses to S and N antigens after dual-antigen hAd5 S + N prime vaccination alone were equivalent to those from previously SARS-CoV-2-infected patients, and in silico prediction models of T-cell epitope HLA binding suggested that T-cell responses to the hAd5 S + N vaccine will retain their efficacy against the B.1.351 variant. Moreover, plasma from previously SARS-CoV-2-infected patients showed a higher binding affinity to cells expressing the dual antigen S-Fusion + N-ETSD construct than to the hAd5 S-Fusion alone, further suggesting that the immunogenicity of the S + N double-antigen vaccine is better than that of the S single-antigen vaccine [26].

The live virus was not detected in the lungs, and weight loss post-challenge was mitigated in the pS, pS + pN, and pS + pE + pM groups, while the pN treatment did not effectively reduce the virus titer. These findings emphasize the indispensability and effectiveness of the S protein as a vaccine target. Notably, co-immunization with pS and pN had better effects than either pS or pN on viral clearance. The pS + pE + pM group indicated fewer histopathological changes in the lungs, which is consistent with the results of our previous study [15]. The S + N group had low viral RNA copies in the lung, reduced weight loss, and a quick recovery time post-SARS-CoV-2 challenge compared to the group immunized with S/N alone, which was consistent with the results of this study. However, none of the groups detected the neutralizing antibody titers, which might be explained by differences in vaccine variety and experimental animals [26]. One SARS-CoV-2 adenovirus vector vaccine study reported that the S vaccine only provided acute brain protection when co-immunized with an N vaccine [27]. Another study developed Tri: ChAd, Bi: ChAd, and Mono: ChAd vaccines expressing the S1/N/RdRp, N/RdRp, and S1 proteins, respectively, and tested them in a B.1.351 animal model. Extensive gross pathology was observed in the Mono: ChAdlungs, whereas Bi: ChAd and Tri: ChAd lungs appeared nearly free of this pathology [28].

Furthermore, unvaccinated animals had high lung viral loads, whereas the Tri: ChAd treatment significantly reduced viral loads by 3.5 logs. In comparison, both Bi: ChAd and Mono: ChAd vaccines only moderately reduced the viral load. These results suggest that the protective effect of the S/N double-antigen vaccine against variants may be better than that of the S single-antigen vaccine, which is consistent with our results [28]. A few studies have reported that N protein-immunized mice develop severe lung inflammation after SARS-CoV infection [29,30,31]. Previous studies have also reported that immunization with an adenovirus vector vaccine expressing the Mouse Hepatitis Virus N protein protects mice against lethal infection, which demonstrates that the N protein could generate a protective effect [32]. Furthermore, the group immunized with the CRT/N DNA vaccine had a significantly reduced viral titer post-challenge, with a vaccinia virus expressing the SARS-CoV N protein [33].

One study on the S protein demonstrated that the combined DNA/protein vaccine induced both humoral and cellular immunity better than the DNA/protein vaccine alone [8]. Vaccines targeting only the S protein have demonstrated reduced effectiveness in protecting against mild-to-moderate COVID-19 caused by emerging variants. The roles of conserved SARS-CoV-2 structural proteins, including the N/E/M proteins, are worthy of attention in vaccine design and applications, as vaccine-induced T-cell responses against conserved epitopes are generally unaffected by mutations. One study reported that SARS-recovered patients (*n* = 23) still possessed long-lasting memory T cells reactive to the SARS-CoV N protein 17 years after the 2003 outbreak, which displayed robust cross-reactivity to the SARS-CoV-2 N protein, further validating the use of the N protein as a cross-protective vaccine target [34]. This study demonstrated that the pS/pN co-immunization was associated with higher nAb responses, better viral clearance, and improved cellular immune responses and may provide better protection after the SARS-CoV-2 challenge compared with pS alone. Furthermore, SARS-CoV-2 variants have been shown to infect many animal species, and human-to-animal transmission has been observed in some wild animals and pets [7]. So, the veterinary SARS-CoV-2 vaccine needs more attention. In addition, nanotechnology may be a powerful tool in optimizing vaccines and is worthy of more attention [2].

This study has several limitations. First, we only observed the DNA vaccine strategy in BALB/c mice, and future studies should assess the immunogenic effects of these vaccine regimens in other animal models. Second, additional research is needed to fully understand the molecular mechanisms underlying the augmented nAb- and S-specific CD8 T-cell responses induced by co-immunization using the S and N proteins and to harness this knowledge to optimize COVID-19 vaccine design. Finally, the function of N protein-specific antibodies deserves further study.

In conclusion, this study evaluated the immune-protective potential of co-immunization with the SARS-CoV-2 S, N, E, and M proteins. Several vaccines targeting only the S protein have a reduced protective effect on the emerging variant strains. Our results will lay a foundation for developing a cross-reactive COVID-19 vaccine to control current and emerging SARS-CoV-2 variants and prevent potential β-coronavirus pandemics.

## Figures and Tables

**Figure 1 vaccines-11-00243-f001:**
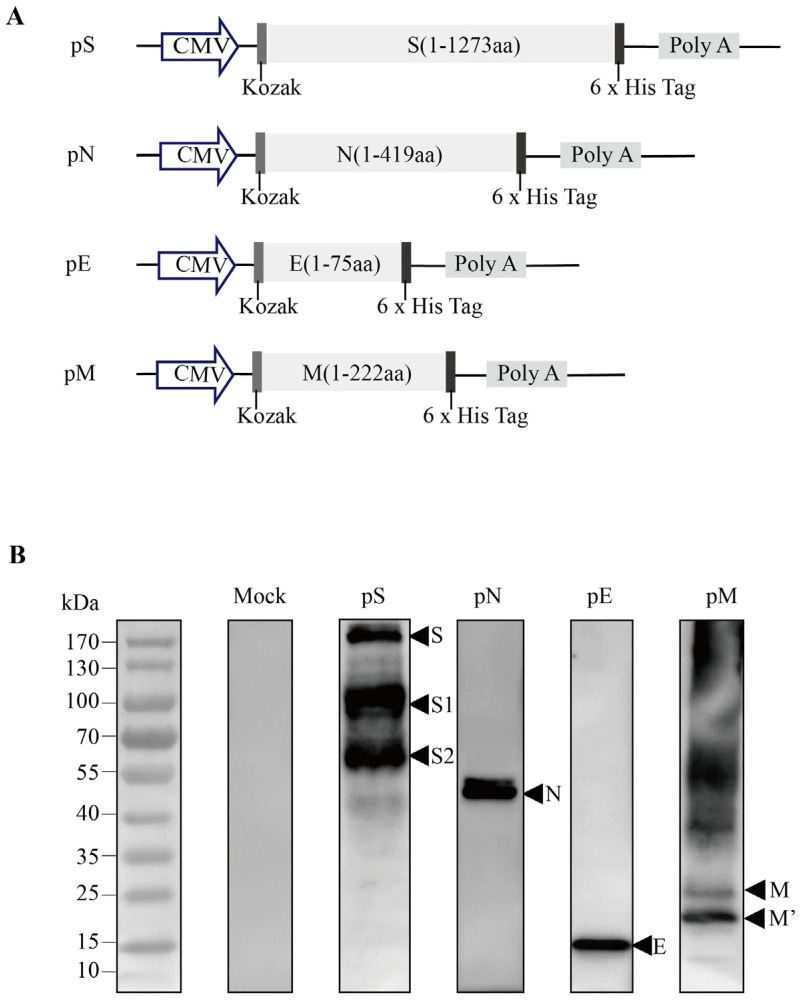
Design and expression of recombinant DNA-based SARS-CoV-2 S/N protein vaccine constructs. (**A**) Schematic diagram of the recombinant DNA-based vaccines encoding SARS-CoV-2 spike (pS), nucleocapsid (pN), envelope (pE), and/or membrane (pM) proteins. (**B**) The target protein expression in DNA vaccines was validated via the western blot analysis of 293T cells transfected with the pS/pN/pE/pM plasmids.

**Figure 2 vaccines-11-00243-f002:**
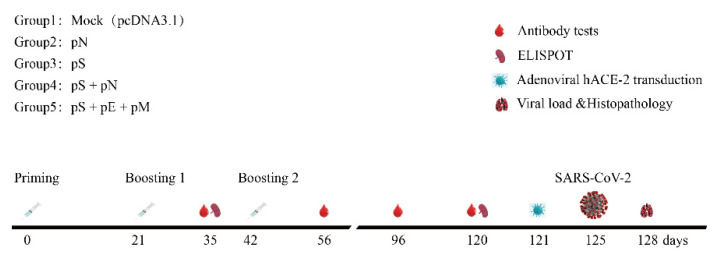
Schematic of the immunization and SARS-CoV-2 challenge. Time course of vaccination, challenging, and blood/tissue sampling. BALB/C mice were divided randomly into groups.

**Figure 3 vaccines-11-00243-f003:**
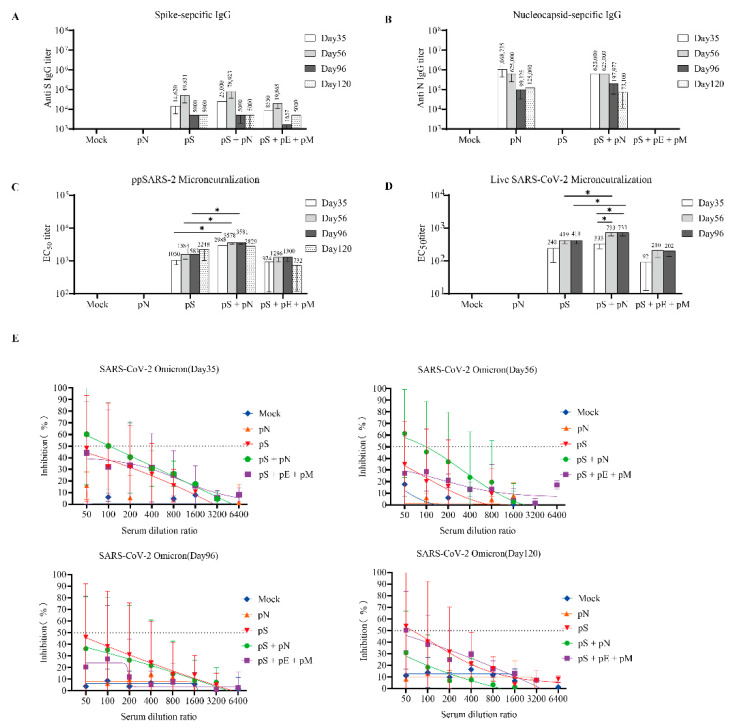
B-cell responses to SARS-CoV-2 in BALB/c mice. (**A**) Serum IgG binding endpoint titers for the SARS-CoV-2 S (**A**) and N proteins (**B**). (**C**) Neutralization titers were determined based on a SARS-CoV-2 pseudotyped-virus system. (**D**) Anti-SARS-CoV-2 neutralization titers were determined using a SARS-CoV-2 virus. (**E**) Neutralization assay based on a SARS-CoV-2 Omicron pseudotyped-virus system. The inhibition ratios for the sera from the mock (blue), pS (red), pS + pN (green), pS + pE + pM (pink), and pN (orange) groups are shown. Error bars represent the SEM, and *p*-values were calculated using a two-way ANOVA and Sidak’s post hoc analysis, where * *p* < 0.05.

**Figure 4 vaccines-11-00243-f004:**
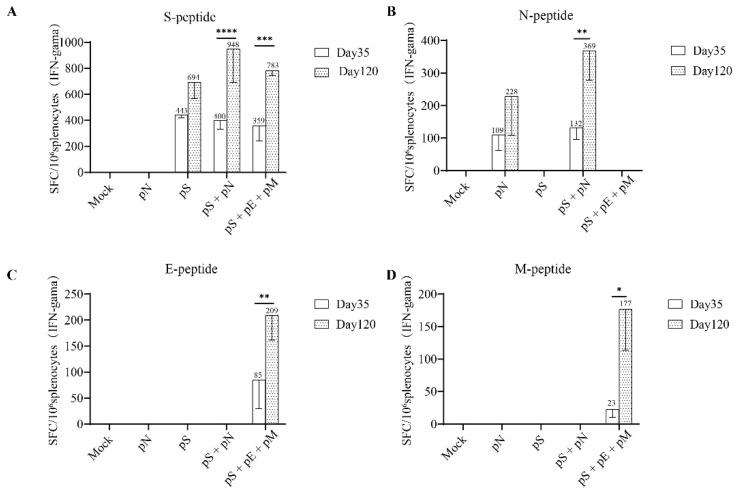
T cell responses to SARS-CoV-2 individual structural proteins in BALB/c mice. (**A**) T-cell responses were measured using IFN-γ ELISpot in splenocytes stimulated for 20 h with overlapping peptide pools spanning the SARS-CoV-2 S, (**B**) N, (**C**) E, and (**D**) M proteins. Bars represent the mean ± SD. Statistical analyses were performed using a two-way ANOVA and Sidak’s post hoc test, where * *p* < 0.05, ** *p* < 0.01, *** *p* < 0.01, and **** *p* < 0.0001.

**Figure 5 vaccines-11-00243-f005:**
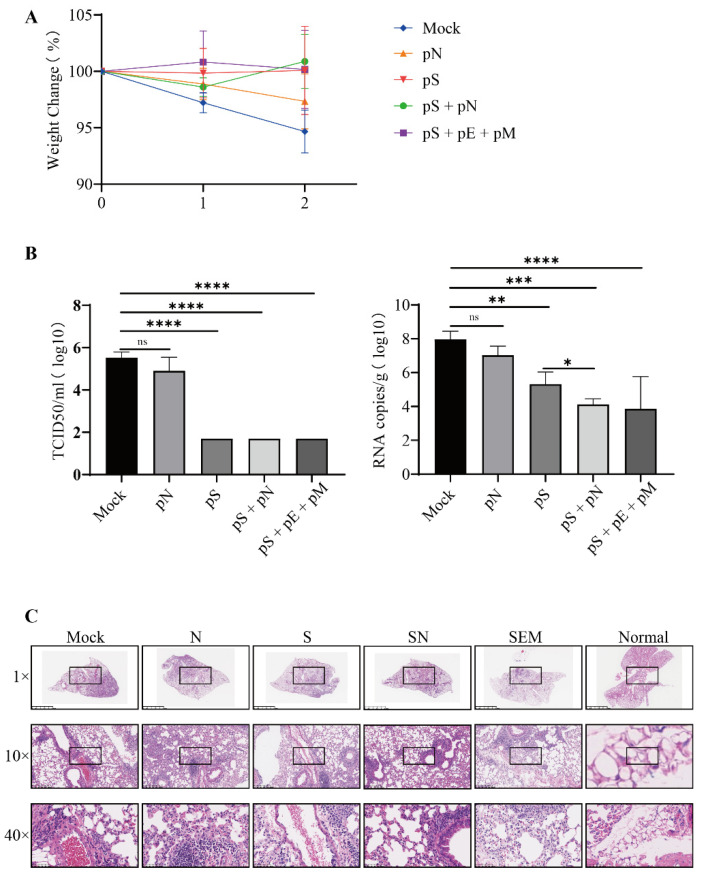
Protective efficacy of immunization after the challenge with live SARS-CoV-2 virus. (**A**) Mice were weighed daily (mean ± standard error of the mean (SEM), *n* = 4) for three days post-challenge. (**B**) Infectious SARS-CoV-2 titer in lung homogenates on day three post-challenge, as determined via the TCID50 assay and RNA copy number. Statistically significant differences between groups were determined using a one-way ANOVA followed by Dunnett’s multiple comparison correction (* *p* < 0.05, ** *p* < 0.01, *** *p* < 0.001, and **** *p* < 0.0001). (**C**) Lung histopathological analysis using H&E staining.

## Data Availability

The data supporting this study’s findings are available on request from the corresponding author.

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
