# Peer review of "Synergistic Immunity and Protection in Mice by Co-Immunization with DNA Vaccines Encoding the Spike Protein and Other Structural Proteins of SARS-CoV-2"

_vaccines, 2023, doi:10.3390/vaccines11020243_

Round 1

Reviewer 1 Report

Jinni Chen and colleagues present a high quality and well-written experimental manuscript reporting synergistic immunity and protection in mice by co-immunization with DNA vaccines encoding spike protein and other structural proteins of SARS-CoV-2.

Authors immunized mice with synthetic DNA vaccines encoding four SARS-CoV-2 structural proteins (pS, pN, pE, and pM) were developed, with three doses via intramuscular injection and electroporation. Notably, co-immunization with two DNA vaccines that expressed the S and N proteins induced higher neutralizing antibodies and was more effective in reducing the SARS-CoV-2 viral load than the S protein alone in mice. In addition, pS co-immunization with either pN or pE+pM induced higher S protein-specific cellular immunity after three immunizations and caused milder histo-pathological changes than pS alone post-challenge. The role of the conserved structural proteins of SARS-CoV-2, including the N/E/M proteins, was investigated further for their applications in vaccine design, such as mRNA vaccines.

Authors argue that this study has several limitations. First, they only observed the DNA vaccine strategy in BALB/c mice, and future studies should assess the immunogenic effects of these vaccine regimens in other animal models. Second, additional research is needed to fully understand the molecular mechanisms underlying the augmented nAb and S-specific CD8 T-cell responses induced by co-immunization using the S and N proteins and to harness this knowledge to optimize COVID-19 vaccine design. Finally, the function of N protein-specific antibodies deserves further study.

Finally, authors conclude that this study evaluated the immunological effects of co-immunization with the SARS-CoV-2 S, N, E, and/or M proteins. Several vaccines targeting only the S protein have demonstrated reduced effectiveness in protecting against COVID-19 caused by various emerging variant strains. Their results will lay a strong foundation for developing a cross-protective COVID-19 vaccine to control current and emerging concern variants and prevent future β-coronavirus pandemics.

Overall, the manuscript is highly valuable for the scientific community and should be accepted for publication.

======================

Other comments to authors:

1) Please check for typos throughout the manuscript.

2) Authors are kindly encouraged to cite the following article that describes novel approaches to target COVID-19 (i.e. based on cell immunotherapies). 

DOI: 10.3390/biomedicines9010059

Author Response

1) Please check for typos throughout the manuscript.

Thank you for the comments. We have carefully revised our manuscript based on your thoughtful suggestions.

2) Authors are kindly encouraged to cite the following article that describes novel approaches to target COVID-19 (i.e. based on cell immunotherapies). 

DOI: 10.3390/biomedicines9010059

Thanks for your suggestion. We have cited this article in the introduction, and please refer to line 51.

Reviewer 2 Report

1-Please add more in vitro and and animals studies in introduction.
2.what is the suggestion of this study for future works?
3.Please discuss the use of this novel natural based nanomedicine with this system.
4.Pleas discuss the role of gap junctions on effect of effect of these vaccines.

5.Please discuss the role of cross talk between mitochondria proteins and spike protein .

6.There are many studies investigating the importance of topic , Please add these references to your discussion part of the manuscript and compare and bold your study novelty :
-DOI:10.1016/j.omtm.2022.12.015

-DOI: 10.2217/nnm-2020-0441

-DOI: 10.1007/s10517-023-05682-9

Author Response

1-Please add more in vitro and and animals studies in introduction.

Thanks for your suggestion. Several references have been provided in the introduction involving in vitro and animal studies. Please refer to line57(refer6-11).

2.what is the suggestion of this study for future works?

The roles of conserved SARS-CoV-2 structural proteins, including the N/E/M proteins, are worthy of attention in vaccine design and application, as vaccine-induced T-cell responses against conserved epitopes are generally unaffected by mutations. In this study,  co-immunization with two DNA vaccines expressing the S and N proteins induced higher neutralizing antibodies and was more effective in reducing the SARS-CoV-2 viral load than the S protein alone in mice. In addition, pS co-immunization with either pN or pE+pM induced higher S protein-specific cellular immunity after three immunizations and caused milder histo-pathological changes than pS alone post-challenge.

3.Please discuss the use of this novel natural based nanomedicine with this system.

As a cutting-edge tool, Nanotechnology should be considered in optimizing vaccine design. Currently, the most promising vaccines for COVID-19 are made of mRNA from surface proteins of SARS-CoV-2 and encapsulated in nanoliposomes with specific physicochemical properties. However, the DNA vaccine is based on a plasmid, and no additional encapsulation technology is needed. The use of this novel nature-based nanomedicine with this system needs further exploration. In addition, we cite your recommended article about nanotechnology in the introduction and discussion. Please refer to line51 xx-xx and lines346-347.

4.Pleas discuss the role of gap junctions on effect of effect of these vaccines.

To the best of our knowledge, the main function of gap junctions is to connect cells so that molecules may pass from one cell to the other, allowing for cell-to-cell communication, and makes it so that molecules can directly enter neighboring cells without having to go through the extracellular fluid surrounding the cells. Regarding DNA vaccine, the target antigens were produced by transcription and translation process after the DNA enters the cells by electroporation technique. Based on your suggestion, the role of gap junctions on the effect of these vaccines and target antigens is considered in future research.

5.Please discuss the role of cross talk between mitochondria proteins and spike protein.

This is an interesting research direction that deserves our further attention. However, the primary aim of this study is to explore the immunological effects of S in combination with other structural proteins. Based on your suggestion, the role of cross-talk between mitochondria and spike proteins should be considered in future research.

6.There are many studies investigating the importance of topic , Please add these references to your discussion part of the manuscript and compare and bold your study novelty :

-DOI:10.1016/j.omtm.2022.12.015

-DOI: 10.2217/nnm-2020-0441

-DOI: 10.1007/s10517-023-05682-9

Thanks for your suggestion. We have added these references to the discussion. Please refer to line51, lines 330-332 and lines 343-347.

Round 2

Reviewer 2 Report

Accept